# Laser Dissimilar Joining of Al7075T6 with Glass-Fiber-Reinforced Polyamide Composite

**Eneko Ukar [1],\*, Jon Iñaki Arrizubieta [1], Mercedes Ferros [2], Maite Andres [2] and Fernando Liebana [2]**

[1] Mechanical Engineering Department, University of the Basque Country UPV/EHU, 48013 Bilbao, Spain; joninaki.arrizubieta@ehu.eus

[2] Tecnalia Research & Innovation, Ed., 700 48160 Derio, Spain; mercedes.ferros@tecnalia.com (M.F.); maite.andres@tecnalia.com (M.A.); fernando.liebana@tecnalia.com (F.L.)

\* Correspondence: eneko.ukar@ehu.eus; Tel.: +34-946-014-905

**Abstract:** Dissimilar joining between metal and composite sheets is usually carried out by mechanical or adhesive joining. Laser dissimilar joining between metal and composite sheets could be an alternative to these methods, as it is a cost-effective and versatile joining technique. Previously, textured metallic and composite parts have been held together and heated with a laser beam while pressure is applied to allow the melted polymer to flow into the cavities of the metal part. The main issue of this process relates to reaching the same joint strength repetitively with appropriate process parameters. In this work, both initial texturing and laser joining parameters are studied for Al 7075-T6 and glass-fiber-reinforced PA6 composite. A groove-based geometry was studied in terms of depth-to-width aspect ratio to find an optimal surface using a nanosecond fiber laser for texturing. Laser joining parameters were also studied with different combinations of surface temperature, heating strategy, pressure, and laser feed rate. The results are relatively good for grooves with aspect ratios from 0.94 to 4.15, with the widths of the grooves being the most critical factor. In terms of joining parameters, surface reference temperature was found to be the most influential parameter. Underheating does not allow correct material flow in textured cavities, while overheating also causes high dispersion in the resulting shear strength. When optimal parameters are applied using correct textures, shear strength values over 26 kN are reached, with a contact area of $35 \times 45$ mm$^2$.

**Keywords:** laser direct joining; laser structuring; metal polymer joint; groove aspect ratio; shear strength

## 1. Introduction

Weight reduction in vehicle structures is essential for reducing $CO_2$ emissions in combustion vehicles and for increasing the efficiency of electric and hybrid propulsion vehicles, where the reduction of the total weight of the structure, together with the development of more efficient batteries, is one of the fundamental factors for increasing autonomy.

The reduction of weight while maintaining structural integrity is only possible using different combinations of materials and thicknesses. In recent years, the use of high and ultra-high yield strength steels in combination with aluminum alloys and thermoplastic matrix composites has become widespread. The need to join materials of different natures that cannot be welded by means of traditional techniques has made the development of new joining techniques necessary, for example using mechanical joints, adhesives, or other alternatives, such as friction spot joining, ultrasonic joining, or laser direct joining [1]. Traditional mechanical rivet joints have evolved and new techniques are now available, such as self-piercing riveting (SPR), clinching, or flow drilling screws (FDS). However, these solutions provide a spot joint in which both sides must be accessible and delamination of the

fibers in composite materials is a common issue. Compared to welding without a joining element, they also have added cost and weight, which are clear drawbacks.

Adhesive bonding allows for wider material combinations than spot welding and mechanical bonding. With adhesives, it is possible to achieve a shear strength of over 30 MPa, where the larger bonding area compared to spot joints avoids stress concentration. However, to achieve the right degree of strength, the adhesives used are usually thermosetting polymer resins that require remarkably long curing times to achieve the right degree of polymerization, which limits their use in mass production applications. On the other hand, the resins used usually generate volatile organic compounds with a significant environmental impact, requiring appropriate facilities and protective equipment [2].

Katayama first presented the laser transmission joining process for the joining of a 2 mm thick PolyEthylene Terephthalate (PET) sheet with 3 mm thick 304 stainless steel using a 807 nm wavelength diode laser. In this process, the materials were positioned in a pressure tool and the laser beam was used to heat the interlayer and melt the thermoplastic. Because of the process in the interface, a layer of $Cr_2O_3$ oxide is generated and both materials are joined. In this case, the PET is a polymer that has practically no radiation absorption at a wavelength of 807 nm. Thus, the laser beam hitting the PET crosses the material to the junction layer, where the stainless steel absorbs the radiation and heats it up, melting the thermoplastic locally on the junction side [2].

Over the last 10 years, the process has evolved. In order to achieve a more solid bond, it has been proven that a previous texturing facilitates the union of both materials, improving the final strength achieved [3–6]. In addition, recent studies have shown that the combination of heating with ultrasound provides a better result in undoing the air bubbles formed in the joint zone [7,8]. The result is even better when combined with a previous texturing of the bonding zone [9].

In relation to the bonding materials, the original process of laser transmission joining requires one of the two materials to be transparent at the wavelength of the laser beam, limiting the process significantly. An alternative for polymer–metal hybrid joints is to heat the metal part directly, so that the joint area interface reaches the melting temperature of the thermoplastic through conduction. In 2013, Jung introduced the process of laser-assisted metal and plastic direct joining (LAMP), also known as laser direct joining, for joining Carbon Fiber Reinforced Polymer (CFRP) and zinc-coated steel [10]. Unlike the laser transmission joining process, in the LAMP process, the heating of the joint face occurs indirectly (i.e., the laser hits the metal part, heating the material). By transmitting the heat by conduction in the metal, the heating of the composite material is achieved. Thus, the energy required to carry out the joint does not depend solely on the melting temperature and degradation of the thermoplastic. In this case, aspects such as the absorptivity of the metal material, conductivity, and even the thickness used are key parameters. In the study presented by Jung in 2013 [10], the influence of both the forward speed and the power was analyzed using a high power diode laser and a $0.6 \times 11$ mm$^2$ laser spot, concluding that a variation of 100 W or 2 mm/s with respect to the optimum parameters of 400 W and 6 mm/s resulted in a reduction in the final strength of more than 40%.

An alternative to reduce the experimentation is the simulation of the thermal field and the correlation of the results of the evolution of the thermal field with the resistance of the final joint. In 2017, Hussein et al. [11] presented a thermal model programmed in ANSYS software for the simulation of the thermal field in the Poly-Methyl Methacrylate (PMMA) joint with stainless steel AISI 304 by means of laser transmission joining and laser direct joining processes. The validation of the model is carried out by comparing the results with an IR camera, but it has some limitations, especially in the laser transmission joining process. In spite of being a first approximation that provided interesting data, aspects such as pressure or existing surface roughness were not taken into account in the modeling of the contact zone, factors that have a direct influence on the thermal conductivity in the jointing interface. There are references in the literature that show that a pretextured finish provides a greater bond strength. In a study presented in 2010 by Hotkamp et al. [3], the relationship between the textured area and total area (SD) was introduced for the first time as a parameter to compare results with different texturing strategies, although in that case the depth of the texture was kept as a constant

factor. In the study published by Rodriguez-Vidal et al. in 2018 [12], results comparing grooves with different Aspect Ratios (AR) obtained with nanosecond (ns) and Continuous Wave(CW) lasers were presented. A higher repeatability was obtained with the pulsed laser compared to the CW due to the material removal mechanism. Despite reaching interesting final conclusions, these works do not show the relevance of parameters such as the influence of the slot width on the final result.

In the present work, a characterization study of Aluminum 7075 T6 was carried out. For this purpose, several tests were conducted to determine the removal rate and to determine the optimum parameters. Subsequently, grooves with different aspect ratios have been engraved and the influence of this parameter on the resulting shear strength was determined.

## 2. Materials and Methods

### 2.1. Joining Materials

The study was carried out on 7075 T6 aluminum in combination with a composite material consisting of a polyamide 6 (PA6) matrix reinforced with glass fiber. The composite material is marketed by the company Bond Laminates under the name Tepex 102 (Brilon, Germany). The thermoplastic matrix of PA6 in combination with 47% glass fiber reinforcement arranged in multiple layers of continuous fiber provides strength and rigidity in a temperature range between −30 °C and 120 °C. The melting temperature is close to 220 °C.

The composition of Al 7075 is shown in Table 1. In Table 2, the main thermal and mechanical properties of Al 7075T6 and Tepex 102 are summarized.

**Table 1.** Chemical composition of Al 7075T6 [13].

| % | Si | Fe | Cu | Mn | Mg | Cr | Zn | Ti | Zr+Ti | Al |
|---|---|---|---|---|---|---|---|---|---|---|
| Min. | 0.05 | | 1.2 | | 2.1 | 0.18 | 5.1 | | | |
| Max. | 50 | 0.5 | 2 | 0.3 | 2.9 | 0.28 | 6.1 | 0.2 | 0.25 | Rest |

**Table 2.** Physical and thermal properties [13].

| Parameter | 7075T6 | TEPEX 102 |
|---|---|---|
| Elastic Modulus (GPa) | 70 | 22.4 |
| Tensile strength, ultimate (MPa) | 572 | 404 |
| Tensile strength, yield (MPa) | 503 | NA |
| Elongation at break (%) | 7.9 | 2.2 |
| Poisson Ratio | 0.32 | 0.17 |
| Shear modulus (GPa) | 26 | 9.57 |
| Shear Strength (MPa) | 331 | NA |
| Melting point (°C) | 477–635 | 220 |
| Thermal conductivity (W·m$^{-1}$·K$^{-1}$) | 130 | 0.28 |
| Specific heat capacity (J·g$^{-1}$ °C$^{-1}$) | 0.960 | 1.8 |
| Hardness | 150 Brinell (HB) | 119 Rockwell (HRR) |
| Coef. of Thermal Expansion (CTE), linear (20–100 °C) (μm·m$^{-1}$·°C$^{-1}$) | 23.4 | 19 |

In the tests, $105 \times 45$ mm$^2$ flat aluminum specimens with a 2 mm thickness were used in combination with $105 \times 45$ mm$^2$ composite specimens with a 2.5 mm thickness. Laser texturing was carried out with a nanosecond pulsed fiber laser from Trumpf (Trumark 5050, Ditzingen, Germany). The experimental study was carried out with a groove-based texturing pattern, which is schematically shown in Figure 1a. The textured aluminum specimens were joined with composite using a 3.1 kW diode laser from Rofin (DL031Q, Hamburg, Germany) guided by an optical fiber. The specimens were fixed in a jointing tool that provided the necessary contact pressure for the thermoplastic material to flow into the pretextured grooves (Figure 1b). Once fixed, the joint was carried out by scanning the

aluminum surface through a window in the jointing tool, holding the tool stationary, and moving the beam with a Fanuc S-10 robotic manipulator (Oshino, Japan) at a constant feed rate.

The resulting textures were measured with a Leica DCM 3D confocal microscope (Wetzlar, Germany) that allowed the reconstruction of the 3D topographies. From the 3D measurements, 2D profiles were extracted using a specific program developed in Matlab © (r2018a) to obtain the average profile with the corresponding deviations.

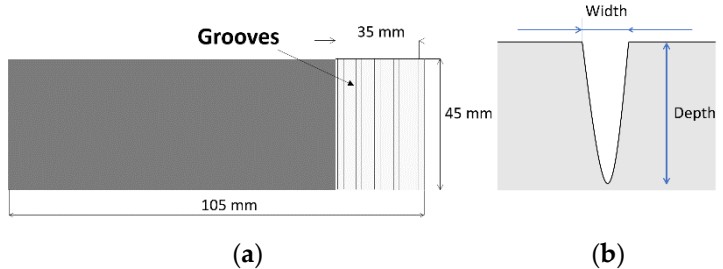

**Figure 1.** (**a**) Textured specimen dimensions and (**b**) textured groove.

## 2.2. Laser Texturing

The texturing was carried out with a Trumark 5050 fiber laser which provides an average power of 50 W and pulses of up to 7 ns, with a maximum frequency of 1 MHz. The laser beam was guided by a scan-head with focal length f = 160 mm and at maximum scanning speeds of 10 m/s, with a spot diameter of 50 μm in a workspace of $110 \times 110$ mm$^2$. In order to homogenize the initial surface and avoid variations in energy absorption, the surface was first scanned at high frequency and speed to prepare the specimens. The parameters used for surface cleaning are shown in Table 3.

**Table 3.** Surface homogenization parameters for Al 7075T6.

| Surface Preparation | Al 7075T6 |
|---|---|
| Number of hatchings | 1 |
| Feed rate (mm/s) | 7 |
| Pulse Frequency (kHz) | 19 |
| Hatching (mm) | 0.05 |
| Pulse duration (ns) | 40 |
| Mean Power (W) | 50 |

To evaluate the material texturing, a surface pattern of grooves was chosen. The results were evaluated in terms of groove depth and width, as well as the final lap shear resistance after joining. The geometry of the grooves allowed evaluation of the material removal rate through the measurements of width and depth in each groove. The Trumark 5050 is a pulsed laser with a nanosecond MOPA-type architecture that generates maximum energy pulses at a frequency of 50 kHz for a duration of 250 ns. In these pulses, the maximum power peak of 10.26 kW is reached in the first 30 ns of the pulse, then decreasing exponentially until the end of the 250 ns phase. With these parameters, the maximum laser energy is released in the shortest pulse time, reaching a maximum pulse energy of 0.93 mJ. For shorter pulses, both the maximum power and the pulse energy are reduced. For pulses longer than 250 ns, the energy released is also 0.93 mJ, but the maximum power is lower as the pulse duration is increased. To achieve high material removal rates, it is necessary to release the maximum energy as fast as possible; for that reason, in the experimental study, the pulse duration was fixed at 250 ns ($t_p$), and taking 50 kHz as the reference frequency, several tests were carried out at a lower frequency of 30 kHz and a higher frequency of 70 kHz. On the other hand, the effect of the feed rate and the number of overlapped tracks on the resulting texture were also studied.

Table 4 shows the test parameters employed for the characterization of the Al 7075T6 texturing, where D is the depth of the grooves, W is the width of the grooves, Ep is the pulse energy, $V_f$ is the scanning speed of the laser beam, Ap is the area swept by the beam in the 250 ns ($t_p$) of pulse, F is the creep, and AR is the aspect ratio between the depth and width of the groove.

**Table 4.** Tested parameters for Al 7075T6. Note: D = depth of the groove; W = width of the grooves; Ep = pulse energy; $V_f$ = scanning speed of the laser beam; Ap = area swept by the beam in the 250 ns ($t_p$) pulse; F = creep; AR = aspect ratio between the depth and width of the groove.

| Frequency (kHz) | Ep (mJ) | $V_f$ (mm/s) | Ap (µm) | N Tracks | W (µm) | D (µm) | F (J/cm²) | AR (D/W) |
|---|---|---|---|---|---|---|---|---|
| 30 | 0.48 | 750 | 0.1972 | 1 | 65 | 15 | 24.33 | 0.231 |
| | | | | 3 | 70 | 35 | 73.34 | 0.5 |
| | | | | 5 | 68 | 60 | 122.23 | 0.882 |
| | | | | 7 | 55 | 80 | 171.12 | 1.455 |
| | | | | 10 | 39 | 110 | 244.46 | 2.821 |
| | | 800 | 0.1973 | 1 | 60 | 10 | 24.32 | 0.167 |
| | | | | 3 | 67 | 28 | 73.34 | 0.418 |
| | | | | 5 | 60 | 53 | 122.23 | 0.883 |
| | | | | 7 | 53 | 77 | 171.12 | 1.453 |
| | | | | 10 | 39 | 104 | 244.46 | 2.667 |
| | | 900 | 0.1974 | 1 | 68 | 8 | 24.30 | 0.118 |
| | | | | 3 | 63 | 20 | 73.34 | 0.317 |
| | | | | 5 | 48 | 43 | 122.23 | 0.896 |
| | | | | 7 | 33 | 60 | 171.12 | 1.818 |
| | | | | 10 | 39 | 90 | 244.46 | 2.308 |
| 50 | 0.93 | 750 | 0.1972 | 1 | 67 | 21 | 47.14 | 0.313 |
| | | | | 3 | 65 | 51 | 141.42 | 0.785 |
| | | | | 5 | 59 | 97 | 235.7 | 1.644 |
| | | | | 7 | 46 | 138 | 329.99 | 3 |
| | | | | 10 | 45 | 165 | 471.41 | 3.667 |
| | | 800 | 0.1973 | 1 | 65 | 19 | 47.13 | 0.292 |
| | | | | 3 | 70 | 45 | 141.38 | 0.643 |
| | | | | 5 | 61 | 80 | 235.63 | 1.311 |
| | | | | 7 | 44 | 129 | 329.88 | 2.932 |
| | | | | 10 | 50 | 163 | 471.26 | 3.26 |
| | | 900 | 0.1974 | 1 | 66 | 15 | 47.1 | 0.227 |
| | | | | 3 | 63 | 39 | 141.29 | 0.619 |
| | | | | 5 | 62 | 75 | 235.48 | 1.21 |
| | | | | 7 | 46 | 116 | 329.67 | 2.522 |
| | | | | 10 | 41 | 157 | 470.96 | 3.829 |
| 70 | 0.637 | 750 | 0.1972 | 1 | 63 | 21 | 32.29 | 0.333 |
| | | | | 3 | 57 | 63 | 96.87 | 1.105 |
| | | | | 5 | 43 | 113 | 161.44 | 2.628 |
| | | | | 7 | 41 | 134 | 226.02 | 3.268 |
| | | | | 10 | 39 | 175 | 322.89 | 4.487 |
| | | 800 | 0.1973 | 1 | 61 | 21 | 32.28 | 0.344 |
| | | | | 3 | 67 | 45 | 96,84 | 0.672 |
| | | | | 5 | 44 | 111 | 161.39 | 2.523 |
| | | | | 7 | 41 | 136 | 225.95 | 3.317 |
| | | | | 10 | 40 | 177 | 322.79 | 4.425 |
| | | 900 | 0.1974 | 1 | 68 | 19 | 32.26 | 0.279 |
| | | | | 3 | 63 | 51 | 96.77 | 0.81 |
| | | | | 5 | 48 | 90 | 161.29 | 1.875 |
| | | | | 7 | 33 | 141 | 225.81 | 4.273 |
| | | | | 10 | 39 | 184 | 322.58 | 4.718 |

The energy density (ED), defined as energy per surface area, depends on the pulse energy at different frequencies, the area swept during the pulse, and the number of repetitions in each area, which is equivalent to the number of overlapped tracks. For the calculation of the area, the spot diameter on the surface is considered to be 50 µm and the distance traveled in each pulse is taken into account (Figure 2). Thus, the energy density or fluence can be calculated through Equation (1).

$$F = \frac{E_p}{A_p} \cdot N^{\circ} \; Tracks \left[ \frac{J}{cm^2} \right] \tag{1}$$

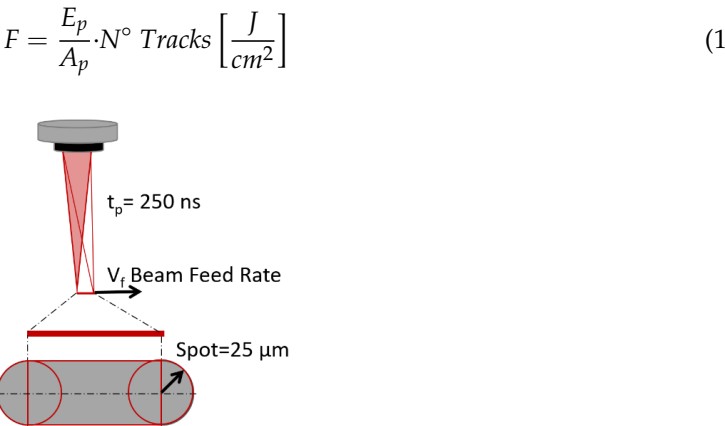

**Figure 2.** Irradiated surface area in each pulse.

## 2.3. Scanning Strategy in Joining

A thermal model was used in order to select the path of the laser beam for joining. The model was based on laser thermal modelling (LATHEM), which is a previous model developed by the University of the Basque Country that was already presented in [14] for LMD process simulation. The model is based on the energy balance schematically shown in Figure 3a. LATHEM was adapted in this work for the application of laser dissimilar joining. Boundary conditions were included to simulate the presence of the composite–polymer part of the joint (Figure 3b). This model is based on the thermal conduction (Equation (2)), which is solved by the finite difference method and was implemented in Matlab®.

$$\alpha \cdot \partial \nabla^2 \theta(x, y, z, t) \mp \frac{q_v}{\rho \cdot c_p} = \frac{\partial \theta(x, y, z, t)}{\partial t} \tag{2}$$

where $\theta$ stands for temperature, $t$ for time, $c_p$ for specific heat, $\rho$ for density, and $\alpha$ for thermal diffusivity. The term $\mp \frac{q_v}{\rho \cdot c_p}$ represents the energy provided to the material. An extended mathematical development of Equation (2) can be found in [15].

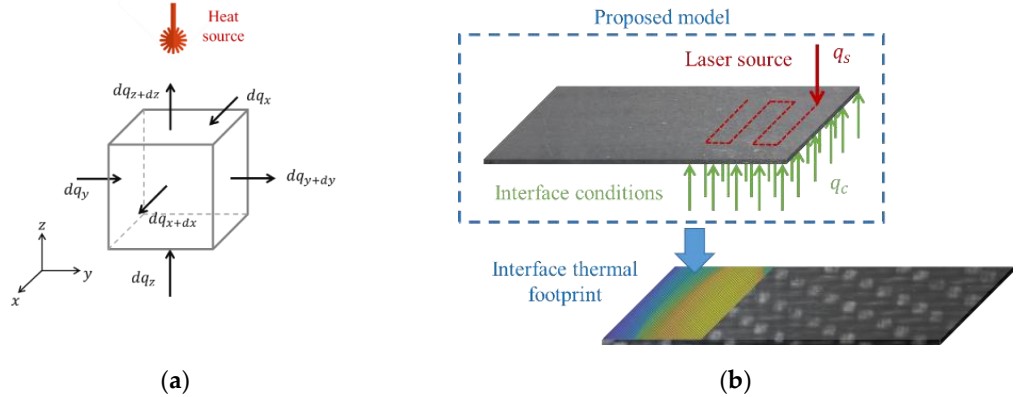

(**a**)                              (**b**)

**Figure 3.** (**a**) Energy balance in each element. (**b**) Description of joining.

The presented formulation is valid for calculating the heat conduction in a body. When two surfaces are in contact, there is a discontinuity in the model that can be modelled as a boundary condition in the area in contact. This heat exchange can be introduced into the model as a boundary condition as well as the heat input. The laser energy is introduced in the model by Equation (3) as a heat flow on the surfaces of the sample, which represents a second-type boundary condition:

$$q_s(x_s, y_s, z_s, t) = -k_a \frac{\partial \theta(x_s, y_s, z_s, t)}{\partial z} = I(x_s, y_s, z_s, t) \tag{3}$$

where $k_a$ is the thermal conductivity of aluminum and $I(x_s, y_s, z_s, t)$ is the intensity of the laser beam in movement through the surface. On the other hand, the contact condition between aluminum and the composite is considered in the model using Equation (4) as a second-type boundary condition:

$$q_c(x_c, y_c, z_c, t) = -k_c \frac{\partial \theta(x_c, y_c, z_c, t)}{\partial z} \tag{4}$$

where $k_c$ is the thermal conductivity of the Tepex 102 composite. Different simulations were carried out to evaluate the influence of the path followed in the thermal footprint at the interface and to find the most efficient trajectory.

The object under evaluation and analysis in these simulations is the homogeneity of the thermal footprint at the interface, in order to find the most efficient trajectory. The model was validated for the 3.1 kW laser used in the experimental joining by measuring temperature data with thermocouples and pyrometers to adjust the energy loss parameter introduced in [16]. The simulations were carried out on aluminum specimens measuring 105 mm in $x$, 45 mm in $y$, and 2 mm in $z$ axes. The mesh discretization was done with values $dx = dy = dz = 0.5$ mm. As in the experimental procedure, the contact area was 35 mm in $x$ and 45 mm in $y$, the trajectory radial step was 5 mm, and margin limits were 2.5 mm in both $x$ and $y$ axes. Three types of trajectories were simulated: zig-zag parallel to the $x$-axis (Figure 4a), zig-zag parallel to the $y$-axis (Figure 4b), and spiral trajectory (Figure 4c).

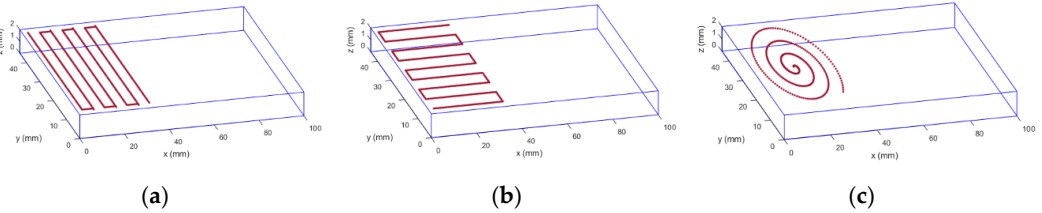

(a)　　　　　　　　　　　　　　(b)　　　　　　　　　　　　　　(c)

**Figure 4.** Simulated joining trajectories: (**a**) parallel to $y$; (**b**) parallel to $x$; (**c**) spiral trajectory.

The temperature range in which the polymeric component of Tepex 102 becomes fluid and where there is no degradation is between 220 and 450 °C, while the minimum time needed to ensure a proper material flow in this temperature range is 2 s. The thermal properties of Al 7075T6 are: conductivity, 130 W/mK; density, 7660 kg/m3; specific heat, 960 J/kgK. The Tepex 102 has a conductivity of 0.28 W/mK. The laser beam source is modelled as a Gaussian distribution with a laser spot diameter of 10 mm. The laser energy loss parameter is 0.2. The parameters under study were evaluated by performing velocity variations of 30–50 mm/s, power variations of 500–3000 W, and 1–2 trajectory repetitions.

### 2.4. Laser Direct Joining

Once specimens were textured, joining parameters were studied. The cell used for laser direct joining had a closed-loop temperature control system through power regulation. Once the reference temperature is established, a pyrometer measures the temperature on the surface and sends a signal that modulates the power to keep this temperature constant. Based on the previous results obtained by Andres et al. [17], different reference temperatures, feed rates, tool closing pressures, and the number of track repetitions were defined. In all the tests, the sweep was kept constant following a zig-zag strategy to keep the temperature distribution as uniform as possible. The aluminum composite specimens were arranged in an overlap configuration so that the laser beam struck the opposite face to the join and the heat reached the interface through conduction. The overlap area of the specimens was $35 \times 45$ mm$^2$. The configuration is shown schematically in Figure 5a. Figure 5b shows an actual picture of the pressure tool used to hold the specimens.

From these simulations, considering only the temperature distribution and the time at which each point is above the melting temperature without reaching degradation, the best results were obtained with a zig-zag trajectory parallel to the *x*-direction. Figure 6 shows the time in seconds at which each point is over the melting temperature of 220 °C and below the degradation temperature of 450 °C for a 50 mm/s feed rate and at 750 W. The area shown in Figure 6 corresponds to the simulation of the joining area and the path followed by the laser is represented in each case. The homogeneity of the thermal footprint is measured as the maximum number of nodes that are in the range of desirable temperatures during a certain period of time. In Figure 6a, the zig-zag trajectory parallel to the *x*-direction provides the most uniform distribution, while Figure 6b shows the result with the same parameters when the zig-zag trajectory parallel to the *y*-axis is followed. In this case, the points on the left side have much greater heat input than ones on the right side, leading to a non-uniform thermal field, which is an undesirable situation. Finally, in Figure 6c the result for a spiral trajectory is shown. In this case, despite the good result reached in the central area of the join, the difference between central and side points is much higher when compared with the zig-zag trajectory parallel to the *x*-axis. Therefore, considering the obtained results, a zig-zag trajectory parallel to the *x*-axis is considered the best option for experimental study.

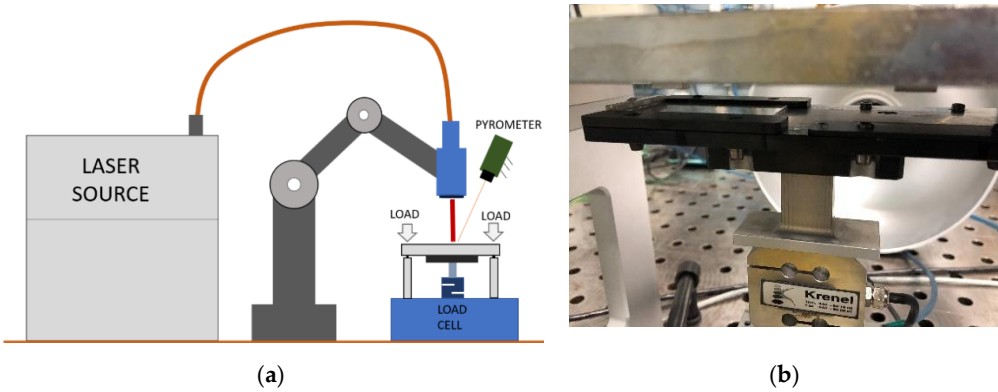

(**a**)    (**b**)

**Figure 5.** (**a**) Joining set-up description. (**b**) Pressure tool details.

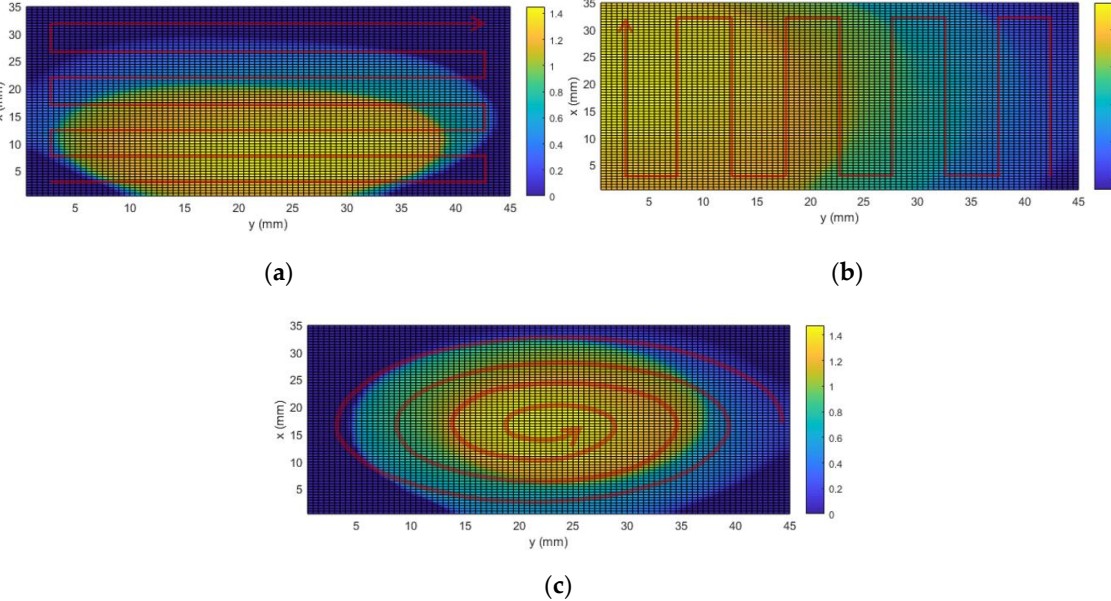

(**a**)    (**b**)

(**c**)

**Figure 6.** The time at which each point is between melting and degradation temperatures: (**a**) zig-zag trajectory parallel to *x*; (**b**) zig-zag trajectory parallel to *y*; (**c**) spiral trajectory.

The different joining parameters studied are shown in Table 5.

**Table 5.** Tested parameters.

| Parameter | Range |
|---|---|
| Temperature (°C) | 340–430 |
| Feed rate (mm/s) | 30–50 |
| Jig Pressure (bar) | 3.5–5 |
| Number of tracks | 1–2 |

Once the specimens were joined, the shear strength was tested by measuring elongation and shear force over time. In order to evaluate the repetitiveness, each test was performed three times. In tests with discrepancy greater than 10%, another specimen was tested and the mean value was taken as a result. Figure 7a shows three specimens joined at 370 °C, 50 mm/s, 3.5 bar, and 2 tracks, which reached the highest shear strength values. Figure 7b shows one of the specimens after the lap shear test, showing homogenous distribution of material in the joining area grooves. The results show maximum values of more than 26.7 kN, equivalent to 17 MPa, for an area of $35 \times 45$ mm$^2$. In total, 32 parameter combinations were tested.

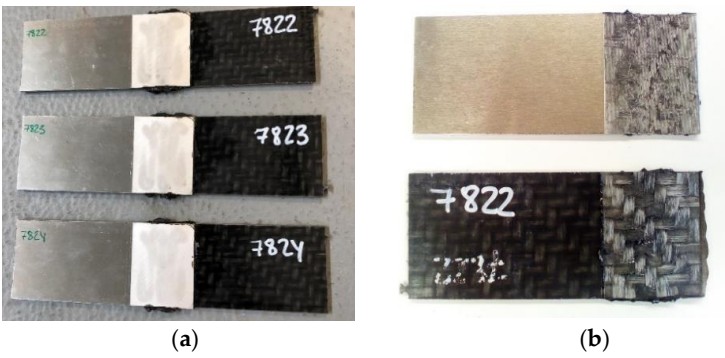

(**a**)                    (**b**)

**Figure 7.** (**a**) Specimens joined at 370 °C, 50 mm/s, 3.5 bar, and 2 tracks. (**b**) The same specimen after lap shear test.

## 3. Results and Discussion

### 3.1. Laser Texturing

From the measurements with the confocal profilometer, the 3D geometry reconstruction was carried out. For a more complete analysis, the geometries were exported as a point cloud to the Matlab$^©$ software for subsequent analysis. Using an own developed algorithm, the profiles of the generated grooves were extracted in each measurement and the average profile with the corresponding deviations was calculated. Figure 8 shows 3D topographies and extracted profiles using Matlab$^©$ for different pulse frequencies using the same feed rate of 800 mm/s and 3 overlapped tracks.

Figure 8a shows the topography reconstruction for 30 kHz and Figure 8b shows the mean profile, where the band containing all extracted profile measurements can also be seen. In Figure 8c,d, the topography and profile measurements for 50 kHz are shown. The extracted profiles show a variability higher than 50 μm in the burr height, causing a non-uniform behavior in shear strength. On the other hand, when the frequency is 70 kHz, Figure 8e,f show better results in terms of burr height homogeneity.

The result was evaluated by measuring the upper width of the grooves, the depth, and the generated slot burr height. In Figure 9, the resulting width and depth for each of the grooves are shown, as well as the aspect ratio (AR) as a function of the energy density (ED). The aspect ratio AR is defined as the ratio between the total depth and the width of the grooves, considering the total depth as the sum of the depth of the slot and the height of the generated burr.

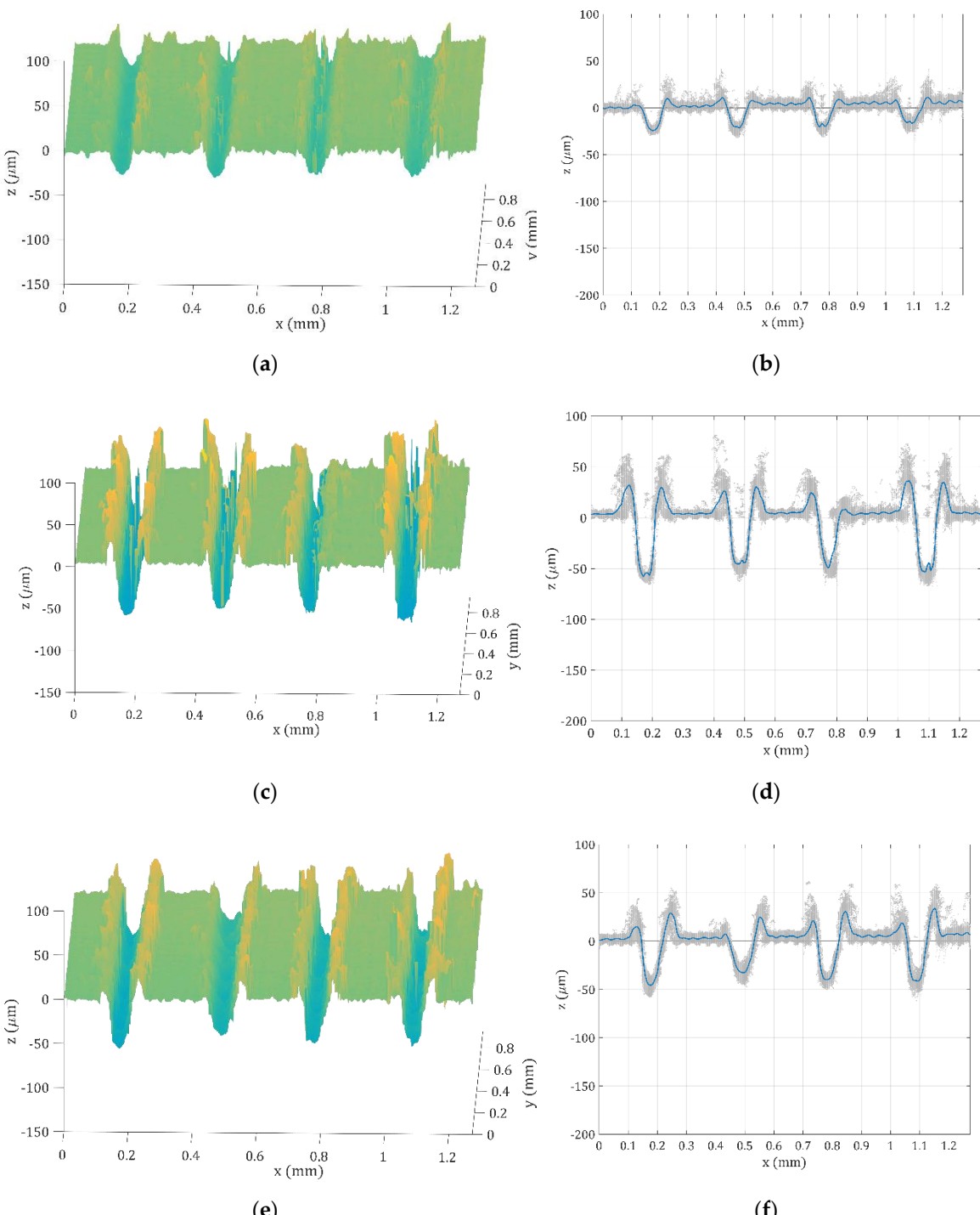

**Figure 8.** (**a**) The 3D topography at 30 kHz. (**b**) The extracted profiles at 30 kHz. (**c**) The 3D topography at 50 kHz. (**d**) The extracted profiles at 50 kHz. (**e**) The 3D topography at 70 kHz. (**f**) The extracted profiles at 70 kHz.

As the number of overlapped tracks increases, the groove depth also increases linearly; however, the evolution of the width is different and as a deeper groove is generated, the upper width of the groove is progressively reduced, as can be appreciated in Figure 9c for 50 kHz and Figure 9e for 70 kHz. This phenomenon is due to the material melting and the vaporization mechanism in the texturing, so during the process, part of the material is melted and is expelled from the cavity, generating an

accumulation of material in the surface area that results in a kind of burr. As the number of overlapped tracks increases, the cavity becomes deeper and some of the melted material does not reach the top, leading to a narrowing of the cavity in the upper area. Since the groove depth is lower with the 30 kHz pulse frequency, this effect is less noticeable (Figure 9a), but also after 10 overlapped tracks the groove width is below the actual laser spot diameter. Thus, when more than 5 or 7 tracks are overlapped, the width of the grooves reach values below the diameter of the laser beam, which is 50 μm at the focal point.

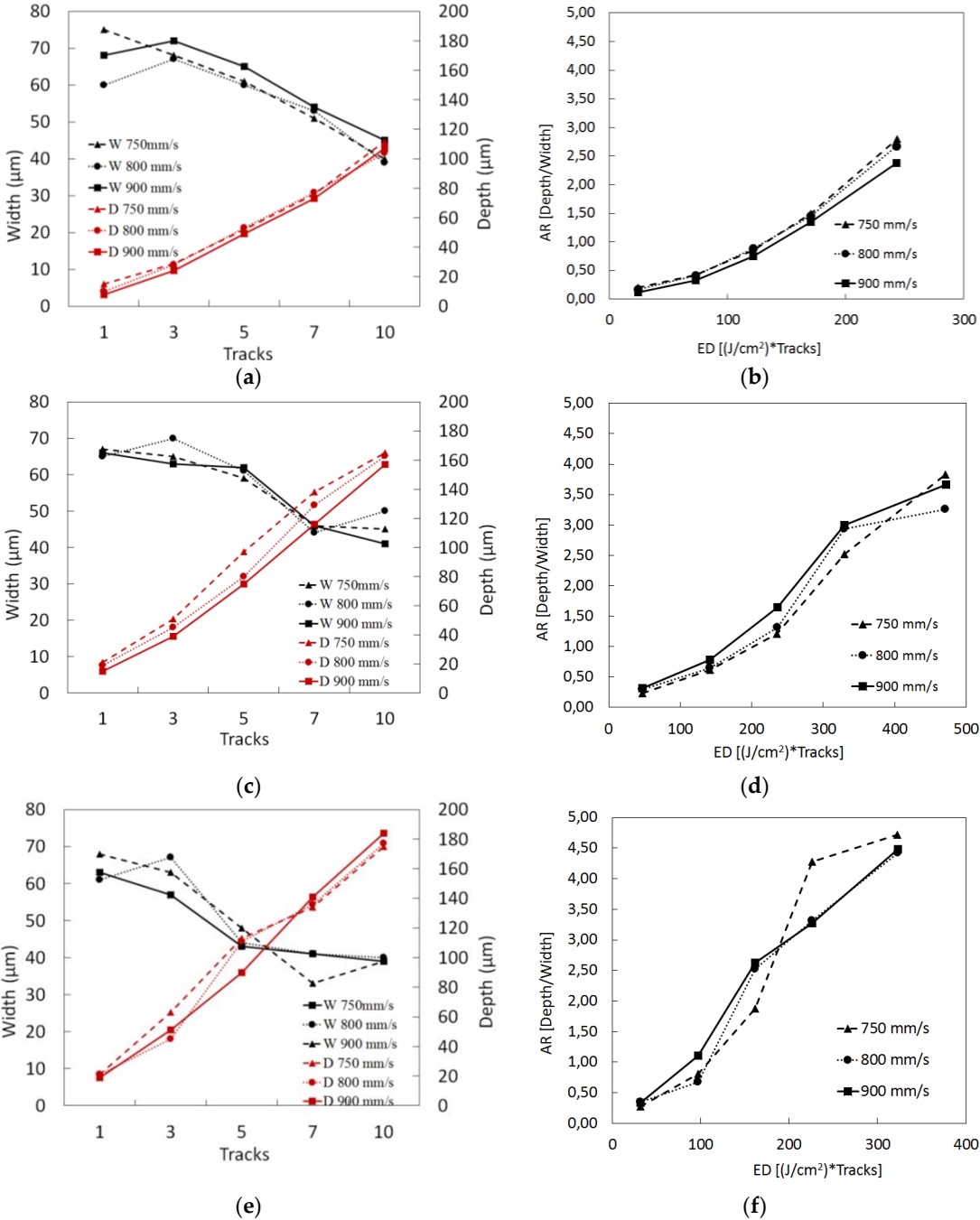

**Figure 9.** (**a**) Width and depth evolution at 30 kHz. (**b**) The Energy Density (ED) influence on Aspect Ratio (AR) at 30 kHz. (**c**) The width and depth evolution at 50 kHz. (**d**) The ED influence on AR at 50 kHz. (**e**) The width and depth evolution at 70 kHz. (**f**) The ED influence on AR at 70 kHz.

On the other hand, the linear behavior in the evolution of depth for the different pulse frequencies and feed rates is remarkable, as can be appreciated in Figure 9a,c,e for different frequencies and feed rates. The slope of the linear approach has little sensitivity to speed variation and increases significantly with an increase of pulse rate. Also, as the frequency increases, a higher material removal rate is achieved, which translates into a greater depth for the same number of overlapped tracks. After 5 overlapping tracks, the slot width decreases to values below the diameter of the laser beam itself, indicating an accumulation of re-solidified material in the upper area of the cavity. If the slot width is below 40 µm, the thermoplastic material cannot flow correctly at the junction. Thus, for 50 kHz the limit would be 5 passes and for 70 kHz the limit would be 3 passes, while at 30 kHz the material removal rate is insufficient and has significant variability. This effect is also noticeable when the groove Aspect Ratio (AR) is evaluated based on the energy density (ED). In Figure 9b,d,f, instead of showing a linear evolution, for ED over 200 J/cm$^2$ the groove exhibits the above-mentioned effect—the width is reduced while the depth is increased, resulting in an AR higher than expected. Taking into account a minimum width of 60 µm, a maximum removal rate, and a minimum deviation for the tests, the parameters that provided better results included a pulse frequency of 70 kHz, a feed rate of 800 mm/s, and 3 overlapped tracks.

With these parameters as a reference, a second study was carried out for the generation of grooves with different AR, but while maintaining a minimum surface width of 60 µm. For this purpose, parameters of 70 kHz, 800 mm/s, and 3 overlapped tracks were used to generate reference grooves. Combining these reference grooves with an overlap ratio of 50%, different AR slots were obtained. Table 6 shows the lap shear results of the slots, with an AR between 0.94 and 4.15 and a minimum width of 60 µm in all cases.

The first joining tests were carried out by taking the joining parameters used in previous work as a reference [16]. These joining tests were carried out to evaluate the influence of texture on the final bond strength. In all the tests carried out, shear strengths above 19 kN were obtained for a joint area of 35 × 45 mm, which is equivalent to 12 MPa. The maximum value was obtained for a slot with two passes in width, where the average value reached was 23.55 kN (15 Mpa). Figure 10 shows the filling of the grooves and the arrangement of the glass fibers in the composite material. A complete filling is observed in all the grooves, which indicates that the joint parameters used ensure the proper flow of the material. On the other hand, an accumulation of passes also ends up generating an area of resolved material that narrows the upper area of the groove, as can be appreciated in Figure 10c,e. In Figure 10a, two reference grooves were overlapped to reach a depth of 102 µm, keeping a width of almost 70 µm in the generated slot. In Figure 10b, as a result of overlapping height and width, the groove generated with the reference parameters is shown. In this way, the depth is almost the same as in Figure 10a, while the slot width is 85 µm. In Figure 10c, two reference grooves are overlapped in depth and three are overlapped in width, with an overlap ratio of 50%. With this strategy, slots of more than 100 µm in width are achieved, also keeping the depth constant at 100 µm. However, in this case the result is much more variable and some slots contain re-solidified material, decreasing the effective width of the slot. In Figure 10d–f, more reference grooves are overlapped in depth while the width is kept constant. In this case, the problem of re-solidified material is more noticeable, giving a higher variability in the results, as can be appreciated in Table 6 and Figure 11, where the mean values of shear strength achieved are summarized for different AR. The results in Figure 11 show that the highest shear strength with the least deviation is achieved with an AR of 1.24, corresponding to the groove shown in Figure 10b.

**Table 6.** Lap shear strength results for different AR grooves.

| Run | Width (μm) | Depth (μm) | Aspect Ratio AR (D/W) | Extension (mm) | Maximum Load (kN) | Mean load (kN) | Standard Deviation (kN) |
|---|---|---|---|---|---|---|---|
| A | 69 | 102 | 1.48 | 2.39<br>1.76<br>2.29<br>1.96<br>3.49 | 17.62<br>16.54<br>17.98<br>21.67<br>24.29 | 19.62 | 3.25 |
| B | 85 | 105 | 1.24 | 2.14<br>2.71<br>2.21<br>2.51<br>2.45 | 22.62<br>23.35<br>24.18<br>24.14<br>23.46 | 23.55 | 0.64 |
| C | 109 | 103 | 0.94 | 2.44<br>1.70<br>2.62<br>2.68<br>2.17 | 23.39<br>18.89<br>23.36<br>22.70<br>20.57 | 21.78 | 1.98 |
| D | 74 | 148 | 2.00 | 2.17<br>2.17<br>2.31<br>1.98<br>2.33 | 18.37<br>22.27<br>24.15<br>20.07<br>21.76 | 21.32 | 2.20 |
| E | 73 | 228 | 3.11 | 2.45<br>1.07<br>1.78<br>1.59<br>2.24 | 25.69<br>13.75<br>19.24<br>18.50<br>20.96 | 19.63 | 4.32 |
| F | 74 | 307 | 4.15 | 3.12<br>2.54<br>2.15<br>2.07<br>2.67 | 23.97<br>24.11<br>20.88<br>21.64<br>19.94 | 22.11 | 1.86 |

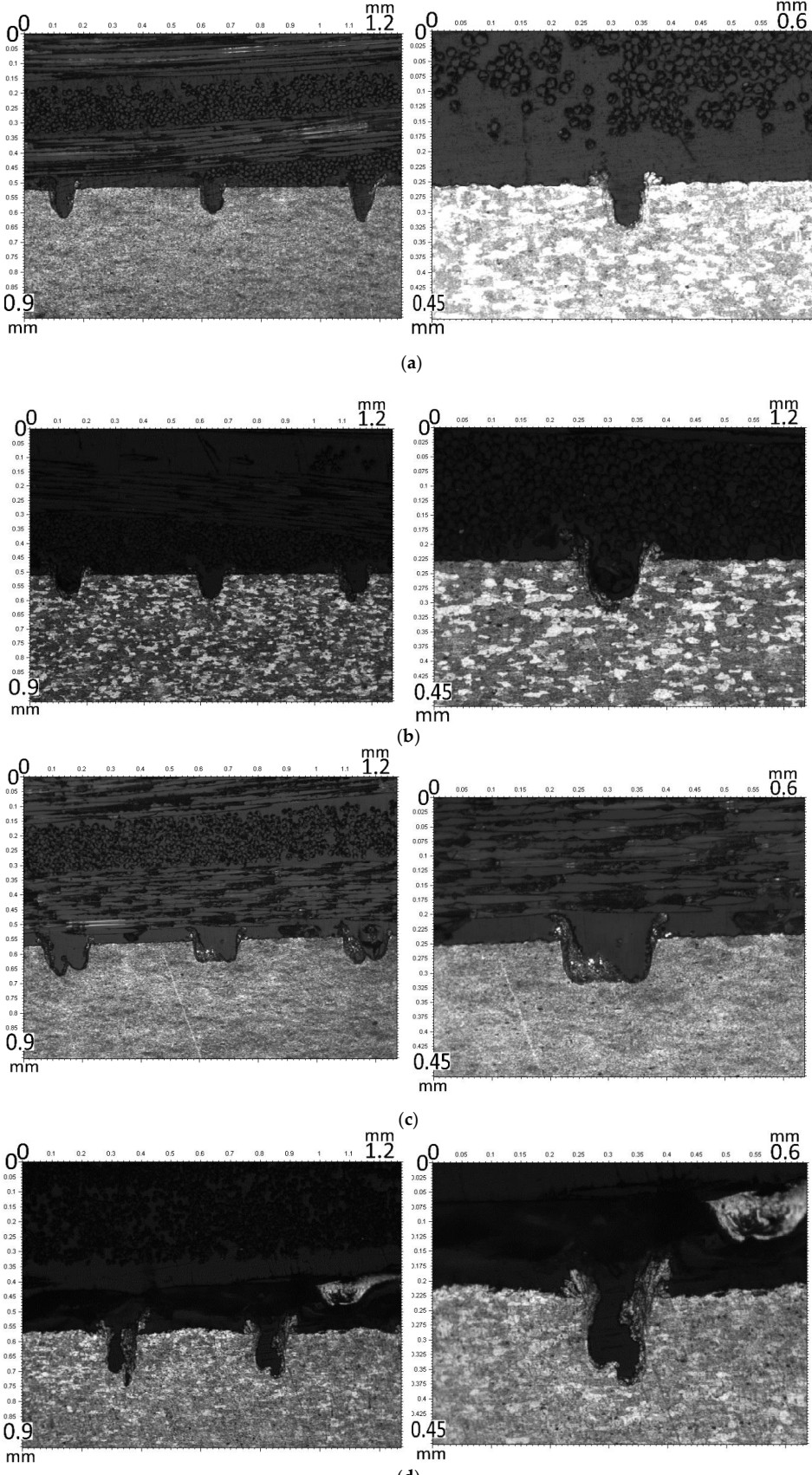

**Figure 10.** *Cont.*

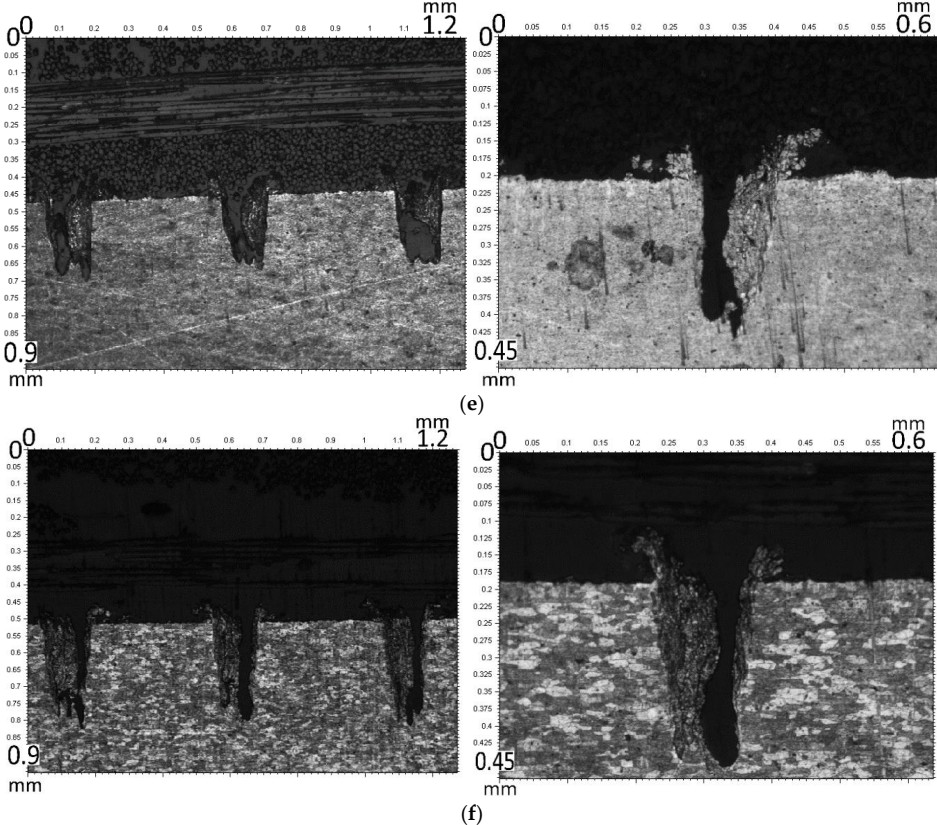

**Figure 10.** Joining results with different AR grooves: (**a**) run A in Table 6; (**b**) run B in Table 6; (**c**) run C in Table 6; (**d**) run D in Table 6; (**e**) run E in Table 6; (**f**) run F in Table 6.

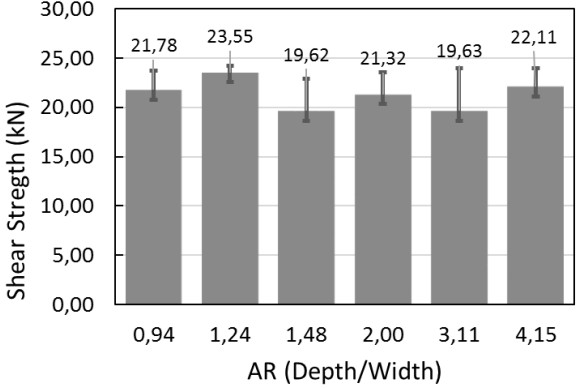

**Figure 11.** Lap shear strength for different AR grooves.

## *3.2. Laser Direct Joining*

All the specimens were tested by measuring the deformation and force according to standard ASTM-D3528 of American Society for Testing and Materials. Joining was carried out following a zig-zag strategy parallel to the shortest dimension with a 5 mm radial step, which is equivalent to a 50% overlap for a 10 mm spot. The process was controlled through a pyrometer, which recorded the temperature of the central point of the $35 \times 45$ mm$^2$ junction area.

The results show that the temperature is the most influential parameter, followed by the feed rate. Figure 12a shows the results obtained for a bonding pressure of 3.5 bar and with two zig-zag sweeps. It can be seen that the set point temperature of the pyrometer is the most decisive factor, with the optimum value being 370 °C. Lower values provide insufficient melting, while higher values cause

excessive melting. Similarly, a lower speed also causes higher heat accumulation, so for the reference temperature of 370 °C, when the feed rate is 30 mm/s, variable results are produced with a significant deviation, as shown in Figure 12a. On the other hand, variables such as tool pressure or even the number of passes have less influence on the result. Figure 12b shows the resistance obtained at a feed rate of 50 mm/s and with two passes for different reference temperatures and tool closing pressures of 3.5 and 5 bar. In this case, the influence of closing pressure is relatively low, with the results being slightly better at 3.5 bar, although in both cases the temperature has a greater influence. Something similar happens when the influence of the number of passes on the process is analyzed. Figure 12c shows the resistance values achieved for a feed rate of 50 mm/s and 3.5 bar pressure when the process is carried out with one and two passes at different set temperatures. For temperatures of 340 °C and 370 °C, better results are obtained with two passes, while for reference temperatures of 400 °C and 430 °C, the result is better with a single pass. This behavior is consistent with the previously identified evolution, in which excessive accumulation of heat was undesirable. Figure 12d shows the results confirming this theory, where a combination of a speed of 30 mm/s and two passes at a pressure of 5 bar gives worse results as the reference temperature increases. The fact that it is more convenient to use a two-pass versus a one-pass sweep strategy doubles the cycle time, however, it should be noted that even if the maximum of 26.7 kN is not reached, it is possible to achieve resistance values above 22 kN by increasing the reference temperature or reducing the feed rate to compensate for the overall energy input.

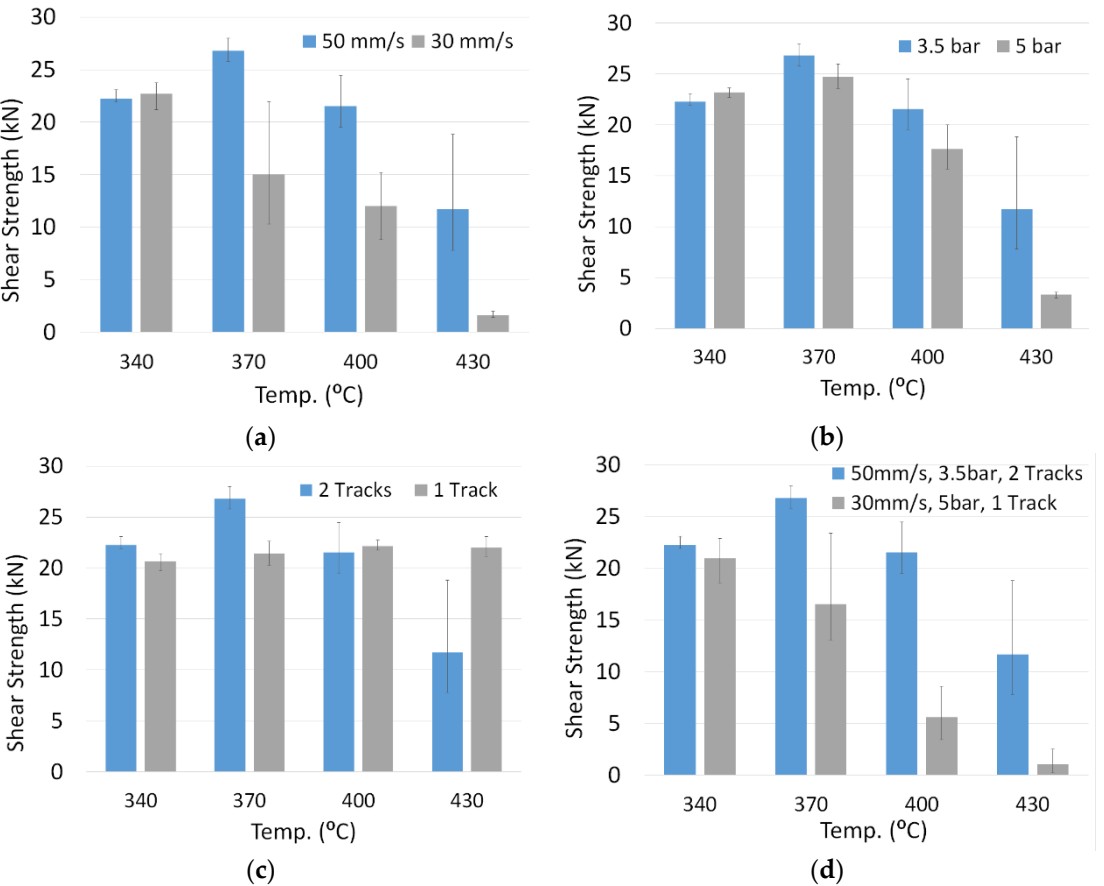

**Figure 12.** Joining parameter influence on lap shear strength: (**a**) feed rate; (**b**) closing pressure; (**c**) overlapped tracks; (**d**) energy input.

In relation to the evolution of the deformation in the shear test, when the joint is adequate, the result obtained is in line with the expected behavior for PA6. Thus, in Figure 13, different curves are observed for joints that reach values ranging from 8 kN to more than 26 kN. Although the evolution of the deformation is different depending on the quality of the joint, in practically all cases a linear evolution can be seen. Taking into account that the total length of the specimen once joined is 175 mm (corresponding to an aluminum and a composite specimen of 105 mm in length minus the overlap area), the maximum deformation achieved for an optimum joint is 6 mm. It should be noted, however, that since the specimen is a multi-material one, it is not possible to define a unitary deformation of the specimen, since in this case the composite's PA6 material matrix inserted in the textured grooves is the one that suffers the deformation.

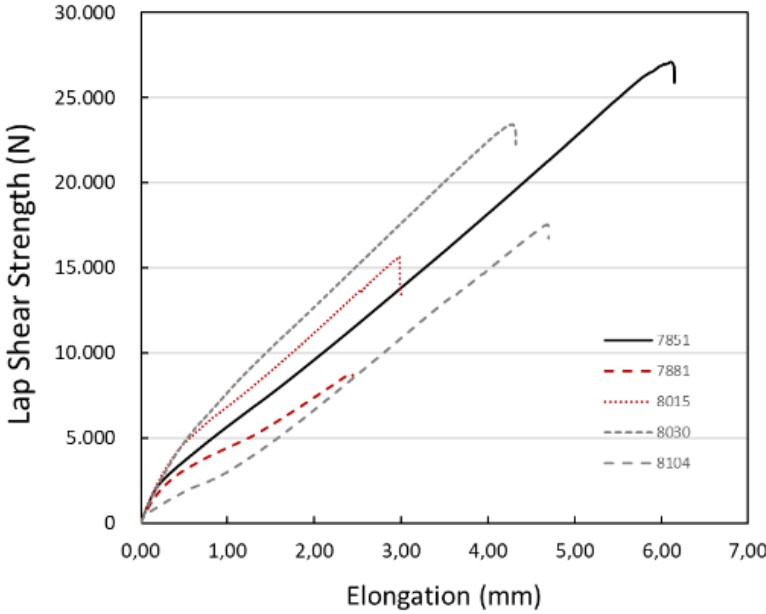

**Figure 13.** Specimen elongation in the lap shear test.

## 4. Conclusions

In this paper, the influences of texturing parameters and bonding parameters on the shear strength of aluminum 7075 T6 joints with Tepex 102 were studied. Groove texturing has been proven to be very effective and allows good adhesion of the materials. The width of the upper groove area is the most relevant parameter in texturing. With a nanosecond laser the material can be effectively removed, but there are certain limits. When grooves are manufactured with several overlapped single tracks, the depth increases in line with the number of overlapped tracks; however, as the depth of the grooves increases, the solidified material tends to accumulate in the upper zone, reducing the width of the grooves. The study carried out at a frequency of 70 kHz provides a more repeatable material removal rate and slots with an aspect ratio of 1.24 provide better results than higher or lower AR. With texturing parameters of 70 kHz, 800 mm/s, and three passes, repeated in a cycle of three passes in depth and two passes in width, a slot of 85 μm in width and 105 μm in depth was achieved, which provided a lap shear strength of over 26 kN and a maximum deformation of 6 mm. It was also seen that a zig-zag strategy parallel to the shorter dimension of the specimen provided a more uniform thermal field. Finally, in relation to the joining parameters, it is concluded that the temperature is the most sensitive parameter. Thus, the feed rate of the beam, the closing pressure, or the number of passes are determining factors, but these have less influence than the temperature. Both a deficit and an excess of energy input into the process results in a less resistant joint. In the study, a reference temperature of 370 °C, a feed speed of 50 mm/s, a tool closing pressure of 3.5 bar, and a two-pass

sweeping strategy provided the best results, giving a shear strength of more than 16.5 MPa for a contact area of $35 \times 45$ mm$^2$.

**Author Contributions:** Conceptualization, E.U. and F.L.; methodology, M.F. and M.A.; data curation, M.F. and M.A.; writing—review and editing E.U. and J.I.A.; supervision, F.L. All authors have read and agreed to the published version of the manuscript.

**Funding:** This research was funded by the Basque Government grant number KK-2017/00088.

**Conflicts of Interest:** The authors declare no conflict of interest.

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
