# Peer review of "Laser Dissimilar Joining of Al7075T6 with Glass-Fiber-Reinforced Polyamide Composite"

_coatings, doi:10.3390/coatings10020096_

Round 1

Reviewer 1 Report

Comments to the author:

1.Please update units correctly to GPa instead of Gpa in table 2. 

2.I suggest that Ap (cm) in table 4 be converted to microns (um) for easy reading instead using E-05 units. 

3.There is descripency in the order and orientation between Figure 4 and 5. Figure 4a has parallel to y, where as Figure 5a has parallel to x and so on. Second, The axis parallel to the view is x in Figure 4, where as y is parallel in Figure 5. This prevents to have a one-to-one correspondence between these two figures at first sight. Please fix the two figures to have uniformity. 

4.Figure 7: Please fully describe the five parts being shown. 

5.Please proofread for multiple errors in English language formatting. 

Author Response

Authors would like to thank reviewers’ effort and time correcting the paper.

Reviewer 1:

Comments to the author:

Please update units correctly to GPa instead of Gpa in table 2. 

Answer:

Units in Table 2 were updated using the correct format.

I suggest that Ap (cm) in table 4 be converted to microns (um) for easy reading instead using E-05 units. 

Answer:

Ap units in table 4 were converted to microns to improve the table.

There is descripency in the order and orientation between Figure 4 and 5. Figure 4a has parallel to y, where as Figure 5a has parallel to x and so on. Second, The axis parallel to the view is x in Figure 4, where as y is parallel in Figure 5. This prevents to have a one-to-one correspondence between these two figures at first sight. Please fix the two figures to have uniformity.

Answer

Figure 5 was modified and path description was included. Additionally, the corresponding explanation was included in the text, where it is detailed that Figure 5 only shows the joining area.

Figure 7: Please fully describe the five parts being shown.

Answer

Explanation paragraph was included in the text with description of Figure 7 and the caption of Figure 7 was corrected. The paragraph included is:

“In order to evaluate also the repetitiveness, each test was performed three times. In tests with discrepancy over 20%, another specimen was tested and mean value taken as result. Figure 7(a) shows three specimens joined at 370 °C, 50 mm/s, 3.5 bar and 2 tracks, which reached the highest shear strength values. Figure 7(b) shows one of the specimens after lap shear test showing homogenous distribution of material in the joining area grooves. The results show maximum values of more than 26.7 kN equivalent to 17 MPa for an area of 35 x 45 mm2. In total 32 parameter combinations were tested.”

Please proofread for multiple errors in English language formatting.

Answer

Article was reviewed and language mistakes were corrected

Reviewer 2 Report

The submitted manuscript entitled ‘Laser dissimilar joining of Al7075T6 with glass fiber reinforced polyamide composite’ deals with the dissimilar joining of a high strength Al alloy and a glass reinforced thermoplastic polymer. The joining is based on the grooves machined into the Al alloy and on the partial melting of the polymer matrix composite. The manuscript well-written and sounds, during its review only a few minor issues (technicalities) as listed below arose.

- Please use ‘×’ instead of ‘x’ in geometrical dimensions.

- The keywords seem to be too general, please detail or select more specified words.

-Table 2: ’MPa’ (‘GPa’) instead of ’Mpa’ (‘Gpa’), please. Do not use dash to divide the denominators (dimension of thermal conductivity, specific heat capacity and CTE). How was the yield strength determined?

Author Response

Authors would like to thank reviewers’ effort and time correcting the paper.

Reviewer 2:

The submitted manuscript entitled ‘Laser dissimilar joining of Al7075T6 with glass fiber reinforced polyamide composite’ deals with the dissimilar joining of a high strength Al alloy and a glass reinforced thermoplastic polymer. The joining is based on the grooves machined into the Al alloy and on the partial melting of the polymer matrix composite. The manuscript well-written and sounds, during its review only a few minor issues (technicalities) as listed below arose.

Please use ‘×’ instead of ‘x’ in geometrical dimensions.

Answer

Multiplication symbol was corrected in geometrical dimensions

The keywords seem to be too general, please detail or select more specified words.

Answer

More specific keywords were used: Laser direct joining; laser structuring; metal-polymer joint; groove aspect ratio, shear strength.

-Table 2: ’MPa’ (‘GPa’) instead of ’Mpa’ (‘Gpa’), please. Do not use dash to divide the denominators (dimension of thermal conductivity, specific heat capacity and CTE). How was the yield strength determined?

Answer

Units in Table 2 were corrected and a new reference was included [13] for composition and mechanical properties details.